# Allelopathic Effects of Caffeic Acid and Its Derivatives on Seed Germination and Growth Competitiveness of Native Plants (*Lantana indica*) and Invasive Plants (*Solidago canadensis*)

Linxuan Pan [1], Feng He [1], Qiuju Liang [1], Yanwen Bo [1], Xin Lin [1], Qaiser Javed [1], Muhammad Saif Ullah [2] and Jianfan Sun [1,3,*]

1  Institute of Environment and Ecology, School of the Environment and Safety Engineering, Jiangsu University, Zhenjiang 212013, China; 2212109030@stmail.ujs.edu.com (L.P.); 2222109001@stmail.ujs.edu.cn (F.H.); LXX2001323@126.com (Q.L.); Yanwen.Bo.627@cranfield.ac.uk (Y.B.); 2222209053@stmail.ujs.edu.cn (X.L.); qjaved@ujs.edu.cn (Q.J.)
2  School of Climate Change and Adaptation, University of Prince Edward Island, Charlottetown, PE C1A 4P3, Canada; msaifullah4704@gmail.com
3  Jiangsu Collaborative Innovation Center of Technology and Material of Water Treatment, Suzhou University of Science and Technology, Suzhou 215009, China
*  Correspondence: zxsjf@ujs.edu.cn; Tel./Fax: +86-511-8879095

**Abstract:** Allelopathy has garnered considerable attention, but the effects of different allelochemicals on invasive plants remain unclear. This study addressed the knowledge gap surrounding allelopathy and its impact on native and invasive plant species. We focused on the impact of caffeic acid and its derivatives on the growth and competitiveness of the native *Lantana indica* and the invasive plant *Solidago canadensis*. We selected three allelochemicals, caffeic acid, methyl caffeic acid, and ethyl caffeic acid, for evaluation at two concentrations (0.1 mM and 1.0 mM). Three planting methods were employed: (1) a single species of *S. canadensis*, (2) a single species of *L. indica*, and (3) a combination of *S. canadensis* and *L. indica*. In addition, a control group was also included. Results revealed that high concentrations (1 mM) of methyl caffeate (MC) and ethyl caffeate (EC) significantly reduced seed germination rate, seed germination index, and seed germination speed index of *L. indica* compared to a low concentration (0.1 mM). Plant height, stem diameter, biomass, and root length in the control group (CK) of *S. canadensis* were significantly higher than those in the treated groups. However, with increasing allelochemical concentration, *L. indica*'s relative competitiveness gradually decreased. These findings provide insights into the concentration-dependent effects of allelopathic compounds on the growth of *L. indica* and *S. canadensis*. By analyzing how these allelochemicals influence the growth and competitiveness of native and invasive plants, the study sheds light on the dynamics of allelochemical interactions between these species. This knowledge can be pivotal for understanding plant competition dynamics in ecosystems and could inform strategies to control invasive species or promote native plant growth.

**Keywords:** allelochemicals; invasive plant; native plant; competition; seedling growth; invasion success





## 1. Introduction

Biological invasion, an integral part of global environmental change, has often been overlooked in previous decades despite its significance. However, with the increasing and diversifying types of invasions, successful invasive alien species now pose direct or indirect threats to biodiversity in invaded areas [1,2]. These invaders severely alter the structure and function of local ecosystems, causing the loss or extinction of native species, leading to overall ecosystem degradation [3]. Additionally, certain invasive species can exploit regional nutritional resources and hinder crop germination and growth, posing substantial risks to regional food security [4,5]. Moreover, specific invasive species with

long flowering periods and abundant pollen production can impose significant health risks on individuals sensitive to pollen, thus affecting daily activities and overall well-being [2]. Despite the growing recognition of invasive species, in-depth research in this field is still needed to investigate and establish the mechanisms underlying the successful invasion of these species.

Allelopathy has garnered significant attention as a key underlying mechanism contributing to the successful invasion of plants [6,7]. Indeed, invasive plants often possess potent allelopathic effects that can significantly impact animals, plants, and soil microorganisms within their colonization range [8,9]. Common allelochemicals are divided into five categories: phenols, terpenes, sugars and glycosides, alkaloids, and non-protein amino acids. These categories encompass various chemical compounds that contribute to the allelopathic effects observed in plants. Each category comprises various specific allelochemicals that exhibit different modes of action and impacts on neighboring organisms [10,11]. By releasing allelochemicals, invasive plants can influence neighboring organisms' behavior, growth, and survival, ultimately promoting their successful invasion [12,13]. Notably, Liu et al. found that plant allelopathic effects can reduce overall plant performance by up to 25% [14].

Overall, the strong allelopathic effects exhibited by invasive plants play a significant role in promoting their invasion by altering the dynamics and interactions within the invaded ecosystem. By impacting plants, animals, and soil microorganisms, invasive plants can reshape community structure, resource availability, and ecological processes, ultimately leading to their successful colonization and dominance in the new environment. As one of the most important plant allelochemicals in the ecosystem, phenolic compounds are widely distributed in plant tissues and rhizosphere soil [15,16]. These compounds significantly affect plant seed development and various metabolic processes through different modes of action. They can modify membrane permeability and affect the uptake of nutrients by plants. They can also inhibit cell division and alter submicroscopic structures, impacting vital processes such as photosynthesis and respiration.

Additionally, phenolic compounds can influence the activities of various enzymes and the synthesis of endogenous plant hormones [3,11,16,17]. Plants release allelochemicals into the environment through different mechanisms, including root exudation, foliar leaching, and decomposition of plant residues. During decomposition, phenolic compounds can be released from leaf litter and other plant remains. Consequently, aerial plant parts can influence the concentrations of phenolic compounds in the soil [18–20]. The accumulation of phenolic compounds in the soil can alter the relationships between the soil and vegetation by affecting the availability of soil nutrients and nutrient cycling [21–24]. Phenolic compounds are important players in allelopathic interactions and can profoundly impact plant growth, nutrient cycling, and soil microbial dynamics. Their release into the environment through various pathways contributes to ecosystems' complex dynamics and functioning [25].

*Solidago canadensis*, a typical alien invasive Compositae plant, exhibits rapid growth, strong reproductive capacity, and a high diffusion rate [26–28]. It can potentially exert allelopathic effects on competing plants [29]. The destructive impact on the natural environment of the introduced area primarily stems from its vigorous growth and robust root system [30]. The content of allelochemicals, including total phenols and total flavonoids, and the allelopathic effects of *S. canadensis*, are typically weaker in their original area compared to the invaded area [31,32]. This suggests that the production and release of allelochemicals by *S. canadensis* may be enhanced in response to the new environment and competing species in the invaded area. Moreover, *S. canadensis* exhibits higher competitiveness in the invaded area, indicating that it has successfully adapted and established itself in the new environment [33,34]. The increased competitiveness of *S. canadensis* can be attributed to various factors, including its ability to produce allelochemicals that inhibit the growth and development of native plants, thereby reducing competition for resources [35].

*S. canadensis*, when invading areas, exhibits strong allelopathic effects that grant it a competitive advantage over native plants, enabling it to dominate invaded ecosystems [36–40]. This demonstrates the critical role of allelopathy in the plant's invasive success, emphasizing the need for strategies to counteract its negative impacts on local biodiversity [41]. While studies have explored the root exudates and rhizosphere soil of *S. canadensis*, research on its essential oils and phenolic components, which play crucial allelopathic roles, is limited [29]. The plant, especially in its polyploid form found in China, enhances phenolic compound metabolism, aiding its invasion [40,42,43]. Notably, Zhang et al. demonstrated that the soil where *S. canadensis* was planted exhibited significantly higher total phenol content than the control soil [44]. Over time, this leads to phenolic acid accumulation in the soil, especially near crop roots, creating a potent microbial environment [44,45].

In this study, we investigated the role of specific compounds extracted from the root exudates of *S. canadensis*, namely caffeic acid, caffeic acid methyl ester, and caffeic acid ethyl ester, to explore the factors contributing to the successful invasion of this species. We raised the following scientific questions: (1) Do root secretion extracts of different concentrations exhibit inhibitory effects on the germination of seeds of the native plant *Lactuca indica* L.? (2) Does the addition of allelochemicals have an inhibitory effect on the growth of native plants, thereby reducing the competitiveness of native species?

## 2. Results

### 2.1. Direct Allelopathic Effects of Phenolic Acid Compounds on Seed Quality, Seed Vigor and Seedling Growth

The quality and viability of seeds are usually evaluated by seed germination rate, seed germination potential, seed vigor index, and seed germination speed. The results demonstrated that compared with the control group, caffeic acid and its derivatives both reduced the quality of the seeds and inhibited the viability of the seeds. The seed germination rate, seed germination index, and seed germination speed index of *L. indica* were significantly affected by the high concentrations (1 mM) of methyl caffeate (MC2) and ethyl caffeate (EC2) treatments, as they exhibited a significant decrease compared to the other treatment groups (Figure 1, Table S1). In contrast, the low concentration (0.1 mM) treatment of methyl caffeate (MC2) and ethyl caffeate (EC2) showed a milder impact on the measured parameters of *L. indica*. Although there was a slight decrease observed in the seed germination rate, seed germination index, and seed germination speed index compared to the control group (CK), the differences were not statistically significant (Figure 1; $p > 0.05$).

Caffeic acid and its derivatives also inhibited the growth of germinated seedlings of *L. indica*. High concentrations of caffeic acid (C2), methyl caffeate (MC2), and ethyl caffeate (EC2) all inhibited the growth of root length, biomass, leaf length, and leaf width of *L. indica* seedlings, but the inhibitory effects of MC2 and EC2 were more significant (Figure 2; $p < 0.05$). In addition, the inhibitory effect of methyl caffeate (MC) and ethyl caffeate (EC) on *L. indica* seedlings was slightly stronger than that of caffeic acid (C) (Figure 2).

### 2.2. Effects on Plant Growth Performance of Invasive Plant S. canadensis and Native Plant L. indica

All three phenolic acid treatments decreased the plant height, root length, and biomass of *S. canadensis* under monoculture (i.e., without interspecific competition) compared with the blank control (CK) (Figure 3, Table S1). The plant height and root length of monoculture *S. canadensis* plants treated with different concentrations of caffeic acid (C1 and C2), caffeic acid methyl ester (MC1 and MC2), and caffeic acid ethyl ester (EC1 and EC2) were significantly lower than those of CK group. It showed a downward trend with the increase in concentration, but there was no significant difference among the treatment groups (Figure 3A,C). The three phenolic acids had no significant effect on the root-to-shoot ratio of *S. canadensis* regardless of whether *S. canadensis* was planted monoculture or mixed. In addition, with the increase in the concentration of the added substances, all three phenolic compounds had obvious inhibitory effects on the biomass of *S. canadensis*

(Figure 3G). The results showed significant differences in *L. indica* plant height and biomass between the control (CK) and treatment groups (except C1 treatment). *L. indica* plants under CK conditions exhibited significantly higher plant height and biomass than those in the treatment group (Figure 3B,H). In addition, with the increase in treatment concentration, the plant height and biomass of *L. indica* showed an obvious concentration–effect relationship.

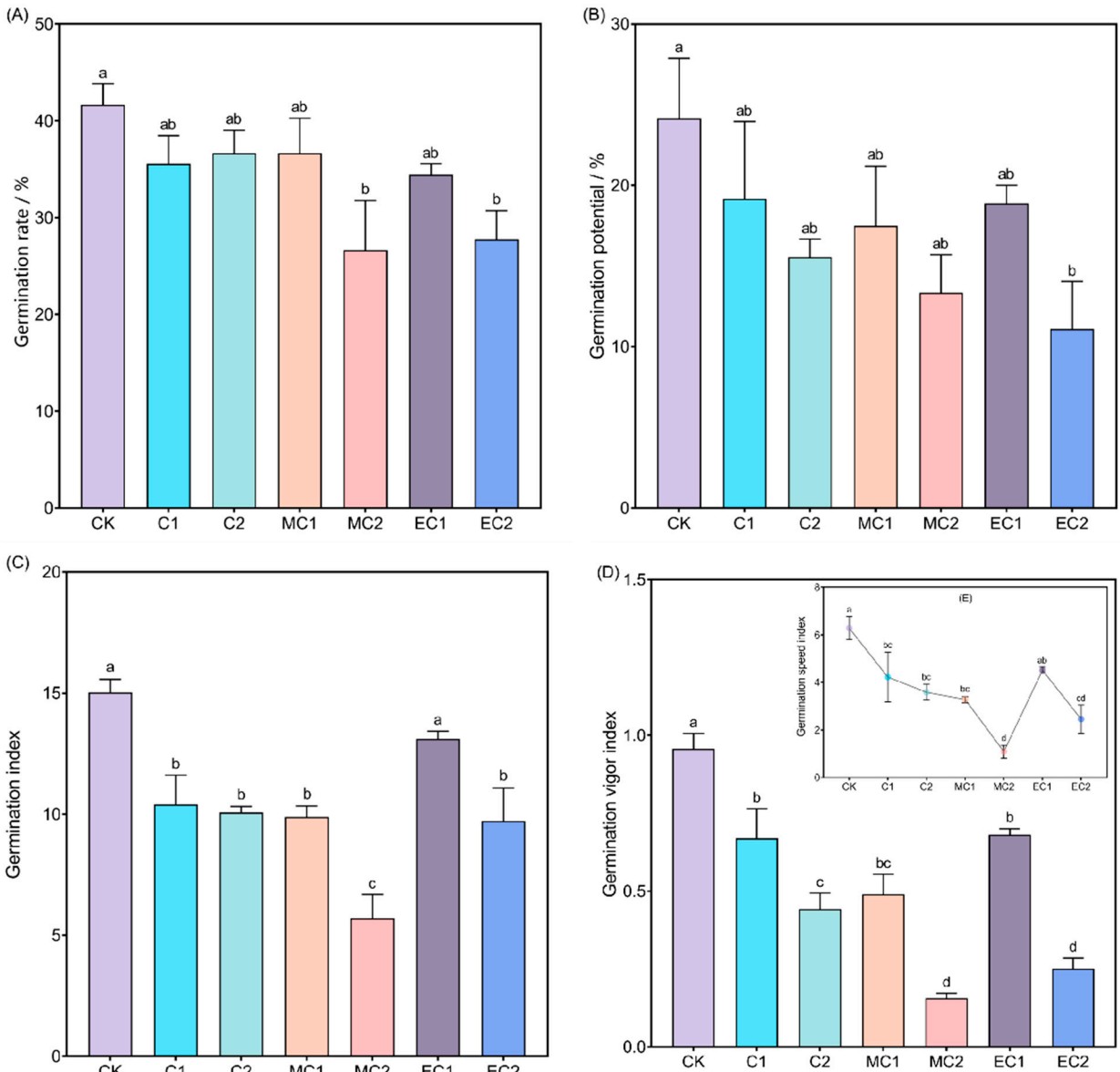

**Figure 1.** Effect of three phenolic acid compounds on (**A**) seed germination rate, (**B**) seed germination potential, (**C**) seed germination index, (**D**) seed vigor index, and (**E**) seed germination speed of *L. indica* at different concentrations. C1 and C2 represent caffeic acid treatment at concentrations of 0.1 mM and 1 mM, respectively; MC1 and MC2 represent treatment with methyl caffeate at concentrations of 0.1 mM and 1 mM, respectively; EC1 and EC2 represent ethyl caffeate at concentrations of 0.1 mM and 1 mM, respectively. Different lowercase letters indicate significant differences ($p < 0.05$).

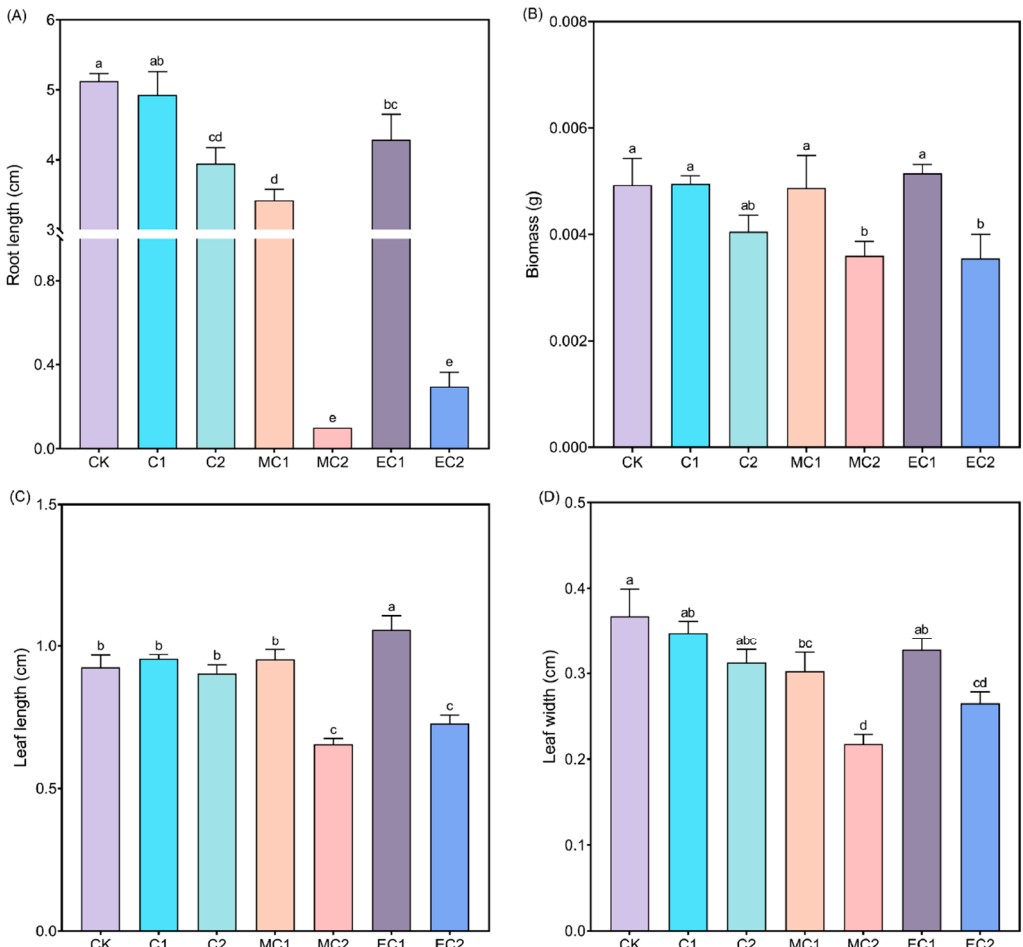

**Figure 2.** Effects of three phenolic acid compounds at different concentrations on (**A**) root length, (**B**) biomass, (**C**) leaf length, and (**D**) leaf width of *L. indica* seeds after germination. C1 and C2 represent caffeic acid treatment at concentrations of 0.1 mM and 1 mM, respectively; MC1 and MC2 represent treatment with methyl caffeate at concentrations of 0.1 mM and 1 mM, respectively; EC1 and EC2 represent ethyl caffeate at concentrations of 0.1 mM and 1 mM, respectively. Different lowercase letters indicate significant differences ($p < 0.05$).

### 2.3. Effects on Plant Physiological Properties of Invasive Plant S. canadensis and Native Plant L. indica

The chlorophyll, leaf nitrogen, and MDA contents of the invasive plant *S. canadensis* had no significant changes under different treatments (Figure 4A,C,E). With the exception EC2, the other five treatments all increased the SOD content compared with CK, but there was no significant difference; although the SOD content of *S. canadensis* plants under mixed planting and mono planting decreased significantly at the two concentrations of EC1 and EC2, respectively, MDA content was not significantly different from that of the CK group, indicating that under this treatment condition, the *S. canadensis* plants had a significant oxidative stress response (Figure 4E,G). For native plant *L. indica*, the chlorophyll and leaf nitrogen content of *L. indica* plants decreased after high-concentration treatment compared with the CK group, among which EC2 treatment was the most significant. In contrast, low-concentration treatment had no significant difference (Figure 4B,D). Each treatment group's MDA content was higher than the CK group's. Still, there was no significant difference among them (Figure 4F). Like *S. canadensis*, the SOD content of *L. indica* plants under mixed planting and mono planting was significantly decreased at two concentrations of EC1 and EC2, respectively. MDA content was not significantly different from that of the CK group's (Figure 4F,H).

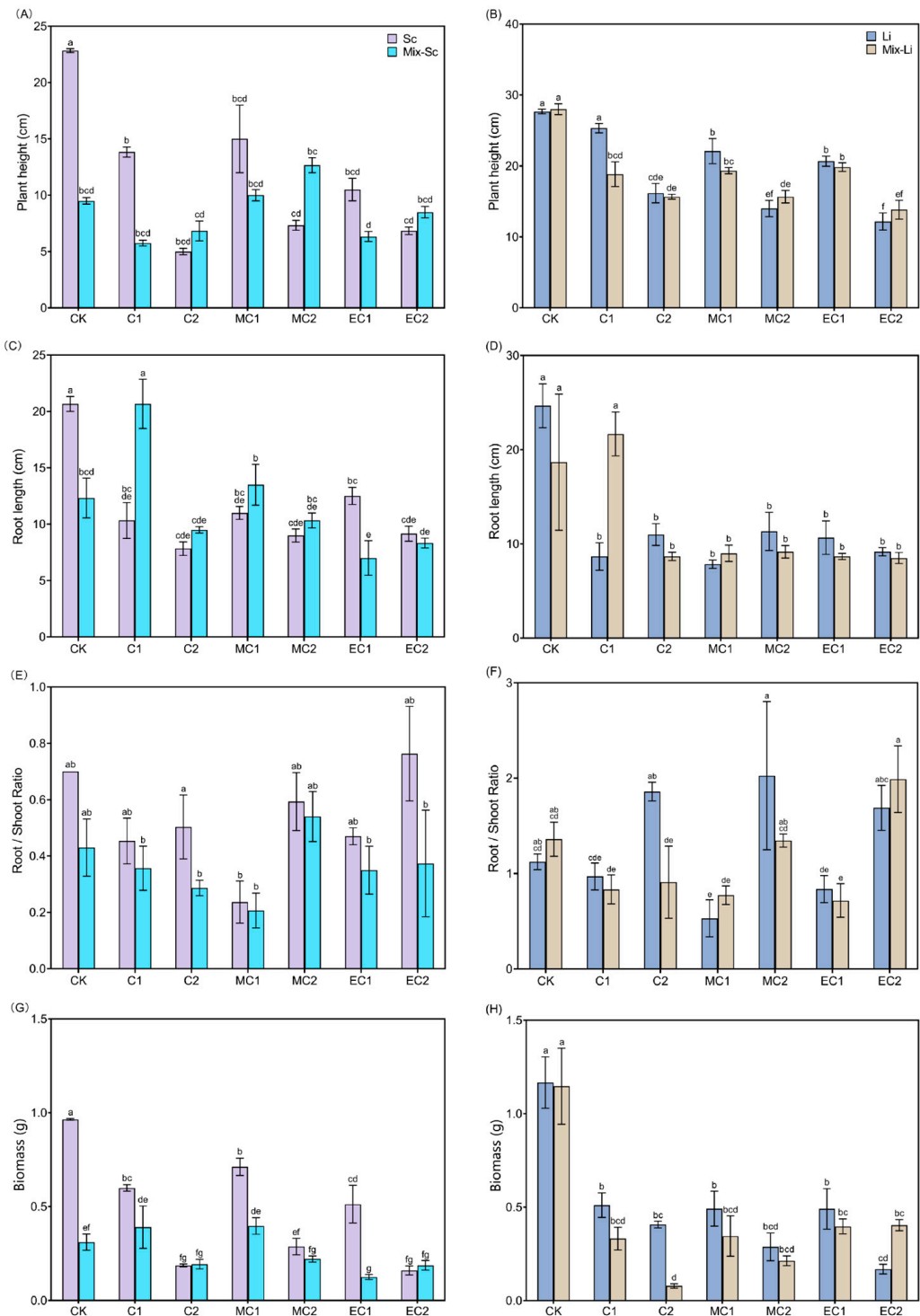

**Figure 3.** Effects of three phenolic acid compounds at different concentrations on the physiological state of *S. canadensis* and *L. indica* plants. Plant height (**A**,**B**); root length (**C**,**D**); root/shoot ratio (**E**,**F**); biomass (**G**,**H**). C1 and C2 represent caffeic acid treatment at concentrations of 0.1 mM and 1 mM, respectively; MC1 and MC2 represent treatment with methyl caffeate at concentrations of 0.1 mM and 1 mM, respectively; EC1 and EC2 represent ethyl caffeate at concentrations of 0.1 mM and 1 mM, respectively. Different lowercase letters indicate significant differences ($p < 0.05$).

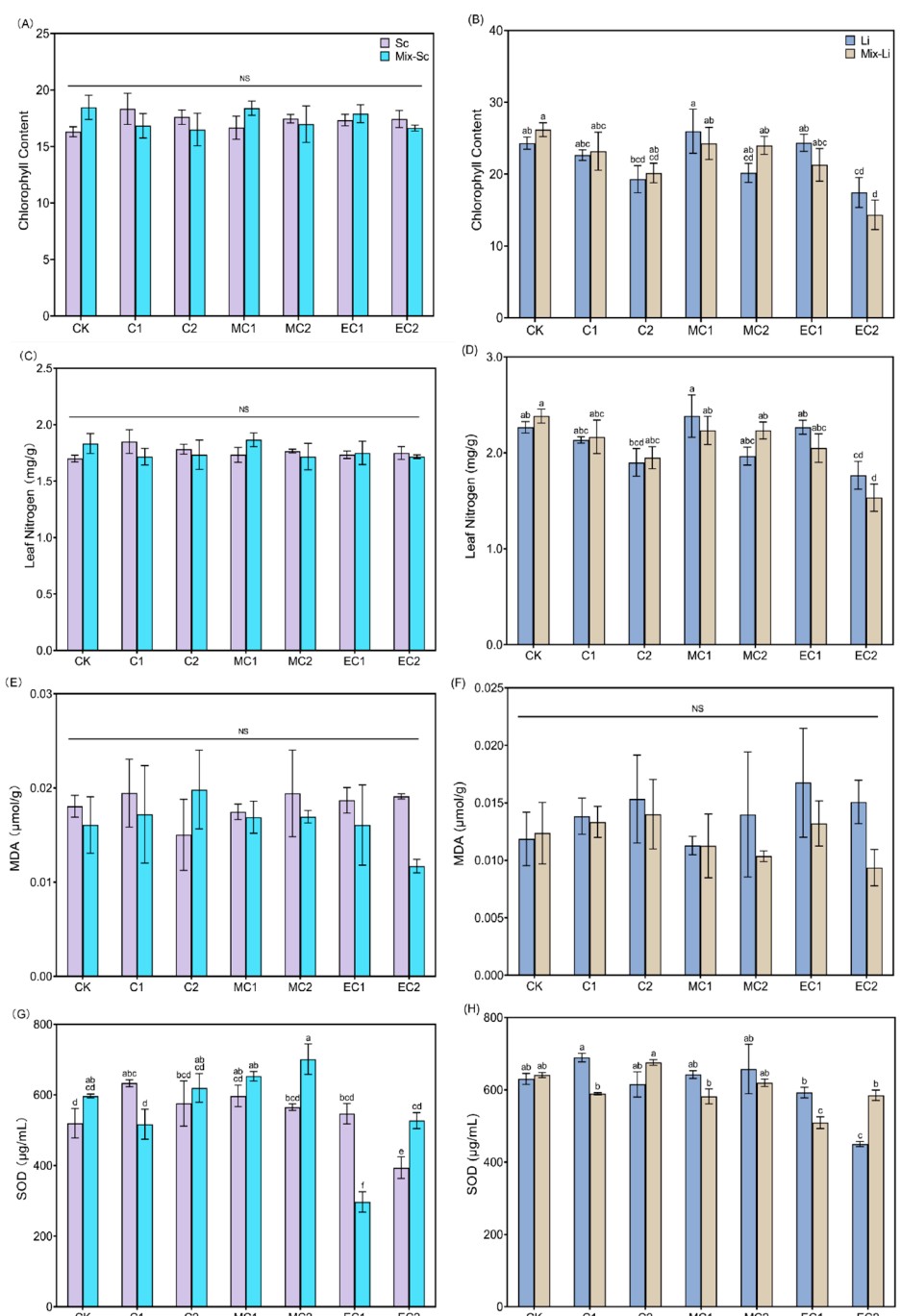

**Figure 4.** Effects of three phenolic acid compounds at different concentrations on the physiological state of *S. canadensis* and *L. indica* plants. Chlorophyll content (**A**,**B**); leaf nitrogen content (**C**,**D**); MDA (**E**,**F**); SOD (**G**,**H**). C1 and C2 represent caffeic acid treatment at concentrations of 0.1 mM and 1 mM, respectively; MC1 and MC2 represent treatment with methyl caffeate at concentrations of 0.1 mM and 1 mM, respectively; EC1 and EC2 represent ethyl caffeate at concentrations of 0.1 mM and 1 mM, respectively. Different lowercase letters indicate significant differences (*p* < 0.05). NS is represented the non-significant.

*2.4. Competitive and Dominance Responses of L. indica and S. canadensis to Varying Concentrations of Phenolic Acid Allelochemicals*

The analysis of the relative competitive intensity index and relative dominance index of *L. indica* and *S. canadensis* under different treatments revealed interesting findings. Under the control conditions (CK), *L. indica* exhibited a higher competitive ability than *S. canaden-*

*sis*. However, as the concentration of allelochemicals increased, the relative competition intensity and relative dominance index of *L. indica* gradually decreased. Notably, when the concentration of allelochemicals reached 1 mM, the growth performance and competitiveness of *L. indica* were significantly inhibited. This inhibition indirectly enhanced the competitiveness of *S. canadensis*, allowing it to exhibit stronger competitive ability in the presence of allelochemicals (Figure 5, Table S2).

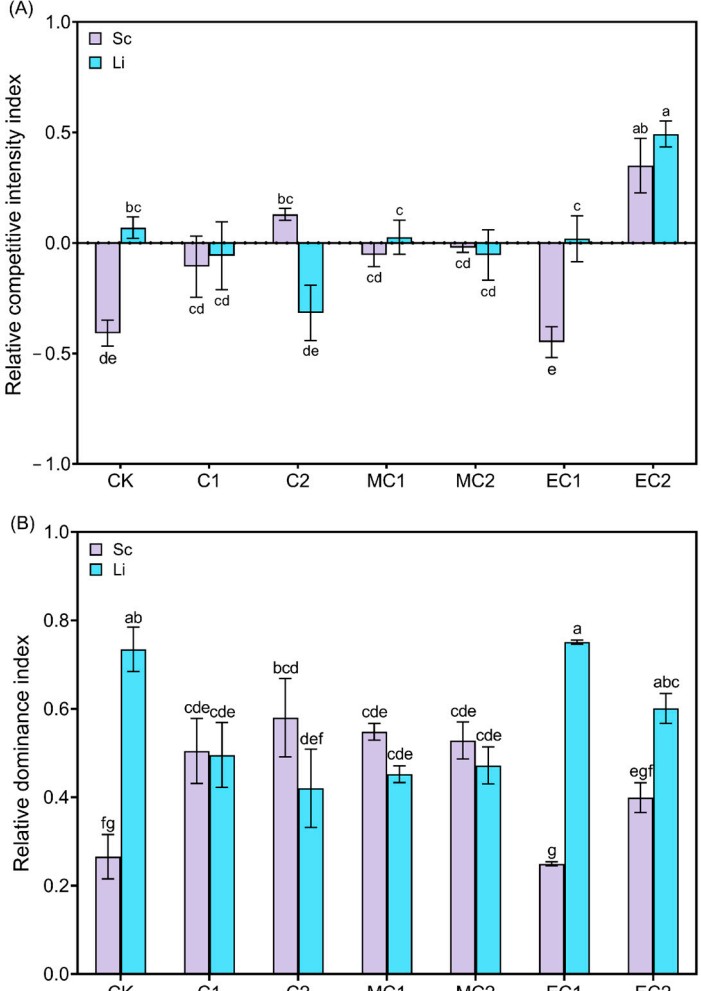

**Figure 5.** Relative competition strength (**A**) and relative dominance index (**B**) of invasive *S. canadensis* and native *L. indica* at different concentrations of three phenolic acid compounds. C1 and C2 represent caffeic acid treatment at concentrations of 0.1 mM and 1 mM, respectively; MC1 and MC2 represent treatment with methyl caffeate at concentrations of 0.1 mM and 1 mM, respectively; EC1 and EC2 represent ethyl caffeate at concentrations of 0.1 mM and 1 mM, respectively. Different lowercase letters indicate significant differences ($p < 0.05$).

## 3. Discussion

### 3.1. Effects of Caffeic Acid and Its Derivatives on Seed Germination

The results obtained from the petri dish experiment indicate that the root length, leaf width, seed germination rate, seed germination index, and seed germination speed index of *L. indica* under the 1 mM concentration of caffeic acid methyl ester (MC2) and 1 mM concentration of caffeic acid ethyl ester (EC2) treatments were significantly lower compared to those of the other treatments (Figure 1, Table S1). These results suggest that these higher concentrations of caffeic acid derivatives had a pronounced inhibitory effect on the growth and germination of *L. indica* seeds. On the other hand, caffeic acid (CA) at concentrations of 1 mM and 0.1 mM had minimal effects on seed germination. This finding is consistent

with Joshi et al. [46], suggesting that caffeic acid alone may have a weak inhibitory effect on seed germination.

It is worth noting that caffeic acid did not significantly affect root length during seed germination. However, caffeic acid significantly inhibited root growth at higher concentrations during the subsequent plant growth stage. This difference in response may be attributed to the long-term accumulation of phenolic acids secreted by invasive plants. The accumulation of these phenolic acids could worsen the growth conditions of native plants, creating a more favorable environment for invasive plants to compete for growth space and resources and promoting their further invasion and colonization [47–49].

These findings highlight the potential role of phenolic acids, such as caffeic acid derivatives, in the allelopathic interactions between invasive and native plants. The inhibitory effects observed on seed germination and root growth suggest that releasing phenolic compounds by invasive plants can contribute to their successful invasion by altering the growth dynamics and competitive advantage in the invaded ecosystems [14,50].

### 3.2. Effects of Caffeic Acid and Its Derivatives on Morphological Indexes of L. indica

Successful invasive species often possess specific traits contributing to their success [51]. Plant height indicates growth and adaptability to stressful environments [1]. In our study, we observed a clear effect of different concentrations of caffeic acid on the plant height of *L. indica* (Figure 1). As the concentrations of methyl caffeic acid (MC) and ethyl caffeic acid (EC) increased, the plant height of *L. indica* and one *S. canadensis* decreased. Interestingly, the plant height of *L. indica* under mixed seed conditions was lower than that under single seed conditions, while the plant height of *S. canadensis* was higher. This finding aligns with the study by Michael O. Adomako et al. [52], which suggests that allelochemicals can enhance the competitive ability of *S. canadensis* and facilitate its invasion by affecting plant growth. Invasive plants can significantly impact the photosynthesis process of native species. High chlorophyll content and strong photosynthesis are favorable for plant growth [53]. The trends observed in chlorophyll content under different concentrations of phenolic acid treatment were consistent with the changes in above-ground biomass and plant height (Figure 2). Additionally, the chlorophyll leaf nitrogen content can indirectly indicate plant photosynthesis levels [53], suggesting that phenolic acid can hinder plant growth by affecting photosynthesis.

Through our investigation of the effects of caffeic acid and other substances on seed germination and plant growth, we found that adding phenolic acid allelopathic substances significantly inhibited the above-ground and underground biomass of *L. indica*. Even at low concentrations, allelochemicals displayed a notable inhibitory effect on the above-ground and underground biomass of *L. indica*. The inhibitory effect of caffeic acid and its derivatives on above-ground biomass was more pronounced at a concentration of 1 mM, and it exceeded the effect on underground biomass. The poor growth of above-ground plants may be attributed to reduced photosynthesis, resulting in reduced competitiveness for water and nutrients against other plants. These findings are consistent with the results of a study conducted by Ren et al. [54].

Furthermore, our results suggest that plants may have the ability to overcome the inhibitory effects of phenolic compounds on the growth of *L. indica*. Biomass serves as a comprehensive reflection of plant growth status, and the addition of allelochemicals primarily inhibits the growth and development of above-ground plant parts, leading to a reduction in photosynthetic efficiency and subsequent growth inhibition. Plant roots play a critical role in nutrient competition [54]. Under stress conditions, plants allocate significant nutrients to rhizosphere growth to withstand stress, resulting in less significant differences in underground biomass. Under the influence of allelochemicals, the above-ground biomass of *L. indica* gradually decreased with increasing substance concentration, indicating a decrease in competitiveness for native *L. indica* with the accumulation of allelochemicals (Figure 2).

### 3.3. Effects of Caffeic Acid and Its Derivatives on Plant Enzyme Activity

Plants have developed endogenous antioxidant enzyme protection systems to defend against oxidative stress induced by various factors, including allelopathic interactions. These systems produce antioxidant enzymes such as superoxide dismutase (SOD), peroxidase (POD), and catalase (CAT) to scavenge reactive oxygen free radicals and protect cellular membranes from peroxidation. Among these enzymes, SOD plays a crucial role in scavenging free radicals in plants [47,48].

The resistance of plants to allelopathic substances is positively correlated with SOD activity [55]. In the presence of low concentrations of allelopathic substances, plants exhibit increased SOD activity, indicating enhanced resistance and alleviation of damage symptoms. The result suggests that plants can adapt and develop a stronger defense response to cope with low-concentration allelopathic stress [56]. However, under high-concentration conditions (1 mM) of allelochemicals, the SOD activity of plants was slightly reduced. This reduction in SOD activity indicates that plants grown under high-concentration irrigation conditions experienced higher stress levels [57]. The negative feedback effect observed on plant growth under high-concentration conditions suggests that caffeic acid and its derivatives harm plant development at elevated concentrations [58–60]. These findings highlight the complex interplay between allelopathic substances, antioxidant enzyme systems, and plant stress response. The balance between allelochemical-induced oxidative stress and the capacity of plants to activate antioxidant defense mechanisms plays a crucial role in determining the outcome of allelopathic interactions [60]. Further research is needed to explore the underlying mechanisms and specific biochemical pathways involved in the modulation of antioxidant enzyme systems and plant responses to allelochemical stress [59,60].

### 3.4. Comparison of Competition between Different Treatments of L. indica and S. canadensis

The study's results revealed important insights into the effects of allelochemicals secreted by *S. canadensis* on the seed germination and plant growth of local plants. The findings demonstrated that caffeic acid methyl ester and caffeic acid ethyl ester had significant effects on seed germination, with their effects increasing as the concentration of these allelochemicals increased.

Moreover, the allelochemicals, including caffeic acid and other substances secreted from the roots of *S. canadensis*, were found to inhibit the growth of native *L. indica* and enhance the competitiveness of *S. canadensis*, especially at higher concentrations. This suggests that the long-term invasion of *S. canadensis* may suppress native *L. indica* growth, creating favorable conditions for the invasion and growth of *S. canadensis*. The competitive advantage of *S. canadensis* can be further enhanced through the accumulation of allelochemicals such as caffeic acid and other substances [40]. Seed germination is a critical process for plant establishment and reproduction [61]. When seed germination is limited, it becomes more challenging for native plants to establish their populations, exacerbating the severity of plant invasion and creating a detrimental cycle [62–64]. The allelopathic inhibition of the root system of *S. canadensis* on the growth of native *L. indica* and the reduction in competitiveness between the two species contribute to the decline of native *L. indica* and the successful invasion of *S. canadensis*.

To effectively counter the ongoing encroachment of *S. canadensis* and mitigate its adverse effects on native plant seed germination, proactive measures involving early detection and eradication are imperative. Complete removal of *S. canadensis* plants, encompassing their root systems, holds paramount importance in thwarting their extensive dissemination and curbing their detrimental impact on the germination process of indigenous plants in the following years. The formulation of efficient management approaches should concentrate on curbing the proliferation and establishment of *S. canadensis* populations, ultimately reducing their ecological footprint within native plant ecosystems and bolstering endeavors aimed at ecological restoration in the agricultural context [65–67].

## 4. Materials and Methods

### 4.1. Seed Collection and Materials Preparation

The seeds of *S. canadensis* and *L. indica* were collected from the outskirts of Zhenjiang City, Jiangsu Province (32°14′26″ N, 119°29′24″ E). Care was taken to select plants with robust growth and similar sizes, and only fully developed and uniformly matured seeds were collected. These seeds were subsequently dried in a cool and dry environment for the experiment. In November 2019, soil samples were collected from Jiaoshan (32°14′26″ N, 119°31′ E), located outside Zhenjiang City, Jiangsu Province, for the pot experiment. The soil of the sampling site is mainly silty clay loam. Soil not inhabited by invasive plants was chosen, and the surface was cleared of local weeds. Approximately 10 cm of the topsoil was collected and dried naturally after passing through an 80-mesh sieve.

Caffeic acid and its derivatives were found in the root exudates of *S. canadensis* in our previous study. Because of the large use of caffeic acid in this study, standard samples were purchased for experiments to ensure the consistency and reliability of the sources of caffeic acid and its derivatives [38]. The allelochemical components used in the study, namely caffeic acid (98%), methyl caffeate (98%), and ethyl caffeate (98%), were obtained from Wuhan Dongkangyuan Technology Co., Ltd., Wuhan, China.

### 4.2. Seed Germination Experiment

Seeds of the same size were subjected to a 15 min immersion in a 0.1% sodium hypochlorite solution to prevent mold formation during germination. Afterward, the seeds were soaked in distilled water for 30 min before being ready for use. A petri dish with a diameter of 90 mm was prepared by placing two layers of filter paper inside. The soaked seeds were evenly distributed on the filter paper in the petri dish, and each allelochemical was added to 5 mL of the two concentration solutions. Each petri dish was equipped with 30 seeds of *L. indica*. The petri dishes were placed in a light incubator at 25 °C for ten days.

For the germination assay, 0.5 mL of deionized water or solutions containing 0.1 mM or 1.0 mM concentrations of caffeic acid, methyl caffeic acid, and ethyl caffeic acid were used. The breakthrough of the seed epidermis at the bud's tip was considered an indicator of germination. The number of germinated seeds was recorded daily. After ten days of cultivation, five seedlings were randomly selected from each petri dish, and growth parameters such as leaf length and width were measured [46].

This study evaluated three allelochemicals, namely caffeic acid, methyl caffeic acid, and ethyl caffeic acid. Two concentrations were tested: 0.1 mM and 1.0 mM. In addition, a blank control was included, where an equal amount of sterile water was added instead of the allelochemical solutions. This resulted in a total of 7 treatment groups (i.e., 3 allelochemical × 2 concentration + 1 sterile water), each with five replicates, thus amounting to a total of 35 petri dish tests.

### 4.3. Pot Experiment

In May 2020, *S. canadensis* and *L. indica* seeds were sown in planting basins. A total of 900 g of field soil was filled in each basin as the test soil. The experimental design consisted of three types of allelochemicals (caffeic acid, methyl caffeic acid, and ethyl caffeic acid) at two concentrations (0.1 mM and 1.0 mM). Each treatment was repeated five times, resulting in 105 potted plants.

Three planting methods were employed: (1) a single species of *S. canadensis*, (2) a single species of *L. indica*, and (3) a combination of *S. canadensis* and *L. indica*. In addition, a control group was also included; that is, no allelochemicals were added, and only the same amount of sterile water was added. Excess plants were removed after the seeds germinated and grew for three weeks, leaving two plants in each pot. On the 15th and 30th of each month, 100 mL of the corresponding concentrations of caffeic acid, methyl caffeic acid, and ethyl caffeic acid were applied as watering treatments. Watering was conducted 2–4 times a week during the remaining time. Two months later, the plants were collected for experimental measurements. Plant leaves were used to measure proline, malondialdehyde,

and enzyme activity. The remaining plant parts were placed in paper envelopes for drying and subsequent weighing to assess biomass or other relevant indicators.

### 4.4. Determination of Physiological and Biochemical Indexes

After ten days of cultivation, five seedlings were randomly selected from each petri dish to measure the germination and growth indices, including plant height, root length, leaf length, and leaf width. The seedling biomass (fresh and dry weight) and allelopathy-related indices were also calculated [68].

Seed germination rate = (normal germination number/total seed number) × 100%

Seed germination potential = (normal germination number/total seed number within 3 d of germination) × 100%

$$\text{Seed germination index} = \sum (\text{Gt}/\text{Dt})$$

where Gt and Dt are the corresponding number of germinating seeds per day and germinating days.

Seed germination rate index = germination index × germination rate

Seed germination vigor index = germination index × seedling fresh weight

The pot experiment measured the relative competitive strength index and relative dominance index for the lower-winged *L. indica* and *S. canadensis* in each treatment group. These indices were used to evaluate the competitive ability of *S. canadensis* under different concentrations of allelochemicals and assess its invasive potential.

Relative competitive strength index measurement [68]:

$$R'_{ij} = \frac{B_{coi} - B_{mi}}{B_{coi} + B_{mi}}$$

where $R'_{ij}$ is the relative competition intensity index of species "i" to species "j"; $B_{coi}$ is the biomass of a single plant when species i and j are mixed; $B_{mi}$ is the biomass of a single plant when species i is single planted.

Determination of comparative advantage index [68]:

$$RDI_{ij} = \frac{B_{coi}}{B_{coij}} \tag{1}$$

where $RDI_{ij}$ is the relative dominance index of species "i" to species "j"; $Bco_{ij}$ is the total biomass of the two species when species "i" and "j" are mixed; $B_{coi}$ is the biomass of species "i" when species "i" and "j" are mixed.

The content of free proline was quantified using the triketone colorimetric method. Malondialdehyde (MDA) levels were determined using the thiobarbituric acid colorimetry technique [2,3]. The activity of superoxide dismutase (SOD) was assessed using the NBT (nitroblue tetrazolium) photoreduction method [69,70].

### 4.5. Statistical Analysis

In this study, we employed analysis of variance (ANOVA) to examine the variations in the effects of different concentrations of three phenolic acid compounds on seed germination and seedling growth of the invasive plant *S. canadensis* and the native plant *L. indica*. To further explore the differences between groups, we conducted the Duncan test to identify homogeneous groups, using a significance level of $p \leq 0.05$. Different letters (a, b, c) indicated significant differences between groups. All statistical analyses were performed using SPSS version 22 (SPSS Inc., Chicago, IL, USA). All figures were plotted using GraphPad Prism version 9.4.1 for Windows (GraphPad Software, Boston, MA, USA).



## 5. Conclusions

The findings of our study exhibit that with the increasing allelochemical concentration, the plant height and root length of *S. canadensis* treated with MC and EC decreased significantly. The SOD content in the leaves of the EC2 treatment group was significantly lower than that in other control groups, suggesting potential impacts on the antioxidant defense system. However, at high concentrations, the growth performance and competitiveness of *L. indica* were significantly inhibited, leading to enhanced competitiveness of *S. canadensis* in the presence of allelochemicals. These findings underscore the concentration-dependent effects of allelopathic compounds on the growth and competitiveness of *L. indica* and *S. canadensis*. Understanding the dynamics of allelochemical interactions between these species is vital for managing and mitigating the impacts of invasive plants on biodiversity. Further research is essential to unravel the underlying mechanisms of allelopathy and its ecological implications.

**Supplementary Materials:** The following supporting information can be downloaded at: https://www.mdpi.com/article/10.3390/agriculture13091719/s1, Table S1: Significance levels of two-way ANOVA for allelochemical type (i.e., C, MC, and EC) and concentration (i.e., 0.1 mM and 1.0 mM) on seed germination and seedling growth performance of the native plant *L. indica*, and their relationship interaction between; Table S2: The significant level of two-factor ANOVA of allelochemicals types (C, MC and EC) and concentrations (0.1 mM and 1.0 mM) on the relative competitive intensity and relative dominance index of invasive plant *S. canadensis* and native plant *L. indica*, and the interaction between them.

**Author Contributions:** Conceptualization, L.P. and J.S.; methodology, Q.J.; software, L.P., Y.B. and F.H.; validation, Q.J., J.S. and Q.L.; formal analysis, X.L., M.S.U. and Q.L.; investigation, Q.J. and J.S.; resources, J.S.; data curation, F.H. and L.P.; writing—original draft preparation, L.P. and Q.J.; writing—review and editing, Q.J. and J.S.; visualization, Y.B.; supervision, J.S.; project administration, J.S.; funding acquisition, J.S. All authors have read and agreed to the published version of the manuscript.

**Funding:** This work was supported by the National Natural Science Foundation of China (31971427, 32071521), Carbon Peak and Carbon Neutrality Technology Innovation Foundation of Jiangsu Province (BK20220030), Priority Academic Program Development of Jiangsu Higher Education Institutions (PAPD), and Jiangsu Collaborative Innovation Center of Technology and Material of Water Treatment.

**Data Availability Statement:** Not applicable.

**Conflicts of Interest:** The authors declare no conflict of interest.

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
