# Peer review of "Allelopathic Effects of Caffeic Acid and Its Derivatives on Seed Germination and Growth Competitiveness of Native Plants (Lantana indica) and Invasive Plants (Solidago canadensis)"

_agriculture, doi:10.3390/agriculture13091719_

Round 1

Reviewer 1 Report

In this article authors tried to understand the role of phenolic compounds on growth and development of invasive plant species. It seems good study in my point of view. Authors did not consistence with their concentration units, like 1µmM (this unit is not correct) and I did not find this concentration in material and methods.  In most of the figures the significance latter are not clear, it needs to be redraw in order to understand properly.    

Minor spell check need to be fixed. 

Author Response

Thank you for your support and valuable comments on this article. This is my mistake regarding the problem that you said the measurement unit is wrong. The concentration of our research is mainly set to 0.1mM and 1.0mM. Thank you for pointing out my mistake. I have double-checked the units of measurement in the text and corrected them. Thanks again for your support and help.

Reviewer 2 Report

Title of the manuscript is suggestive for the content of the study. In the abstract, the purpose (aim of the work) is clearly identified, which is important in my opinion. I think that the justification for the research (its importance) should be specified more clearly and included in the abstract.

The study is complex and quite well organized, fact that shows the correct and detailed documentation of the authors. The conclusions are thoroughly supported by the results.

There are some errors, such as: - line 58  Notably, a study by Liu et al. found that plant allelopathic effects can reduce overall plant performance by up to 25%[14], I suggest without by.

 I recommend a short linguistic check.

Author Response

Reply: Thank you for your support and valuable comments on this article. Your suggestion on the abstract part has benefited me a lot. This study addressed the knowledge gap surrounding allelopathy and its impact on native and invasive plant species. We focused on the impact of caffeic acid and its derivatives on the growth and competitiveness of the native Lantana indica and the invasive plant Solidago canadensis. I will supplement the research purpose and significance of the article in the abstract part according to your suggestion.

There are some errors, such as: - line 58  Notably, a study by Liu et al. found that plant allelopathic effects can reduce overall plant performance by up to 25%[14], I suggest without by.

Reply: Based on your suggestion, we have modified it to: Liu et al. found that plant allelopathic effects can reduce overall plant performance by up to 25%[14]

Reviewer 3 Report

The study was centred on finding allelochemicals for reducing the growth of invasive weed species. The aim was good, the experiments were planned well. However, the there are many casual mistakes in the manuscript. Authors need to re-read the manuscript and correct it. Some of the mistakes are pointed out here-

L14: Spelling mistake on very first word of the abstract

Carefully check the entire manuscript for such mistakes

L18 what is μmM, if you mean micro molar then write μM

L18 and L23 are contradictory, be careful about the unit, one place it is mentioned in micro moles and one place it is milli moles

Use uniform expression throughout the MS, some places μM was used and in some place

Introduction section is too long, it can be reduced

How the statistical analysis has been done, have you used factorial design, if not then the data should be analysed in factorial design

In graphs, mention the units in brackets; e.g. Root length (cm)

Figure 1 and 2 data are only for L. indica, why these factors were not studied for S. canadensis

L377: What do you mean by “appropriate solution? Please mention in the text

L379-380- Correct the unit expression, it is not correct. In half of the manuscript it is “0.1 mM and 1 mM”, and in another half it is “0.1 μmM and 1.0 μmM”, both are not same, there is huge difference

L397-398- Make italics of scientific manes, correct in other part of MS too

Author Response

L14: Spelling mistake on very first word of the abstract. Carefully check the entire manuscript for such mistakes.

Reply: Thank you for your support and valuable comments on this article. The misspelling of the word has been changed to "Allelopathy." We have also removed all the language errors throughout the manuscript.

L18 and L23 are contradictory, be careful about the unit, one place it is mentioned in micro moles and one place it is milli moles. Use uniform expression throughout the MS, some places μM was used and in some place.

Reply: This is my mistake regarding the problem that you said the measurement unit is wrong. The concentration of our research is mainly set to 0.1mM and 1.0mM. Thank you for pointing out my mistake. I have double-checked the units of measurement in the text and corrected them. Thanks again for your support and help.

Introduction section is too long, it can be reduced

Reply: We have simplified part of the introduction according to your suggestion.

How the statistical analysis has been done, have you used factorial design, if not then the data should be analysed in factorial design

Reply: Thanks for your questions and suggestions. In this paper, one-way analysis of variance was used to analyze the data. We will seriously consider your suggestion and consider including factorial design in our data analysis and include it in the supplementary filet.

In graphs, mention the units in brackets; e.g. Root length (cm)

Reply: We have modified the graphs to show it more clearly and beautifully.

Figure 1 and 2 data are only for L. indica, why these factors were not studied for S. canadensis

Reply: The decision to focus solely on L. indica in Figures 1 and 2 was based on our specific research objective: investigate the allelopathic effects of caffeic acid and its derivatives on the seed germination and growth competitiveness of native plants. Since L. indica is a native species of particular interest in our study, we aimed to highlight its response to different treatment conditions.

While we acknowledge that data for S. canadensis would provide a comprehensive view, we intended to delve deeply into the interactions of allelopathic substances with L. indica. We believe that concentrating on one species can provide a more detailed and nuanced understanding of the mechanisms at play.

However, we recognize the importance of investigating the effects on S. canadensis. In subsequent research, we plan to explore the allelopathic effects on S. canadensis and incorporate those findings into a more comprehensive analysis. This will allow us to provide a holistic view of the interactions between allelopathy and native and invasive plant species.

L377: What do you mean by “appropriate solution? Please mention in the text

Reply: We have re-explained it in the article and changed it to: “The soaked seeds were evenly distributed on the filter paper in the petri dish, and each allelochemical was added to 5 mL of the two concentration solutions.”

L397-398- Make italics of scientific manes, correct in other part of MS too

Reply: Thanks for your careful review; we have revised it correctly.